# EEG Evoked Potentials to Repetitive Transcranial Magnetic Stimulation in Normal Volunteers: Inhibitory TMS EEG Evoked Potentials

**DOI:** 10.3390/s22051762

**Published:** 2022-02-24

**Authors:** Jing Zhou, Adam Fogarty, Kristina Pfeifer, Jordan Seliger, Robert S. Fisher

**Affiliations:** Department of Neurology and Neurological Sciences, Stanford University School of Medicine, Stanford, CA 94304, USA; zhou5@g.clemson.edu (J.Z.); afogarty@stanfordhealthcare.org (A.F.); kpfeifer@stanford.edu (K.P.); jseliger@stanford.edu (J.S.)

**Keywords:** transcranial magnetic stimulation, epilepsy, cerebral cortex stimulation, electromagnetic influence, neurostimulation

## Abstract

The impact of repetitive magnetic stimulation (rTMS) on cortex varies with stimulation parameters, so it would be useful to develop a biomarker to rapidly judge effects on cortical activity, including regions other than motor cortex. This study evaluated rTMS-evoked EEG potentials (TEP) after 1 Hz of motor cortex stimulation. New features are controls for baseline amplitude and comparison to control groups of sham stimulation. We delivered 200 test pulses at 0.20 Hz before and after 1500 treatment pulses at 1 Hz. Sequences comprised AAA = active stimulation with the same coil for test–treat–test phases (*n* = 22); PPP = realistic placebo coil stimulation for all three phases (*n* = 10); and APA = active coil stimulation for tests and placebo coil stimulation for treatment (*n* = 15). Signal processing displayed the evoked EEG waveforms, and peaks were measured by software. ANCOVA was used to measure differences in TEP peak amplitudes in post-rTMS trials while controlling for pre-rTMS TEP peak amplitude. Post hoc analysis showed reduced P60 amplitude in the active (AAA) rTMS group versus the placebo (APA) group. The N100 peak showed a treatment effect compared to the placebo groups, but no pairwise post hoc differences. N40 showed a trend toward increase. Changes were seen in widespread EEG leads, mostly ipsilaterally. TMS-evoked EEG potentials showed reduction of the P60 peak and increase of the N100 peak, both possibly reflecting increased slow inhibition after 1 Hz of rTMS. TMS-EEG may be a useful biomarker to assay brain excitability at a seizure focus and elsewhere, but individual responses are highly variable, and the difficulty of distinguishing merged peaks complicates interpretation.

## 1. Introduction

Transcranial magnetic stimulation (TMS) [1,2], repetitive transcranial magnetic stimulation (rTMS) [3,4], and intermittent or continuous theta burst stimulation [5] have been evaluated for therapeutic effects in numerous clinical conditions. Results vary and the optimal parameters of stimulation remain uncertain. For example, of seven controlled studies of rTMS as a treatment for epilepsy, two have been favorable for seizure improvement [6,7] and five documented transient, little, or no benefit against seizures [8,9,10,11,12]. Systematic testing of various stimulation protocols against different clinical outcomes is a lengthy and difficult process. Therefore, a biomarker able to efficiently assay the biological effect of rTMS would likely accelerate development of useful therapies. 

Small variations of stimulation parameters or locations can lead to widely varying–sometimes opposite–clinical responses [13]. For example, low-frequency stimulation at 0.5–1 Hz depresses motor cortex excitability [14], whereas 5 Hz of stimulation increases excitability [15].

The most commonly used biomarker for effects of rTMS is the electromyogram (EMG)-evoked response in the hand, while stimulating contralateral motor cortex [16]. A sufficiently strong TMS stimulation delivered to motor cortex elicits a thumb or finger twitch and an EMG response recorded by a surface electrode on the hand. Cortical stimulation produces local excitation, followed by a silent period reflecting cortical inhibition [17]. A second TMS pulse delivered during the period of inhibition will produce a smaller EMG response in the hand, thereby allowing the ratio of EMG amplitudes of the second versus the first response to serve as a marker of induced cortical inhibition [16]. This method of estimating cortical inhibition only applies to motor cortex. However, the desired inhibitory effect of rTMS often is on another region of the brain, for example, the dorsolateral frontal cortex for treating depression [18] or a cortical seizure focus for treating epilepsy [19,20].

TMS evokes an electrical response in cortex that can be recorded by electroencephalography (EEG) electrodes [21,22,23,24,25,26,27,28,29,30,31]. Because the EEG signals are low amplitude and distorted by magnetic pulse artifact, signal averaging of multiple stimuli and digital signal processing methods are required to characterize the EEG response to TMS [32,33,34,35,36,37,38,39,40,41]. Nevertheless, TMS-evoked EEG potentials (TEPs) can be assessed in any region of cortex before and after a putative therapeutic maneuver. This potentially affords an opportunity to use TEPs as a biomarker of rTMS or TBS efficacy. Changes in TEPs in response to rTMS treatment were demonstrated by the present authors in a single case study of a patient with epilepsy, also correlating with improvement in seizures [42].

The usual TEP has negative (N) and positive (P) peaks at N20, P30, N40 [43] (sometimes labeled as N45), P60 (sometimes labeled P70), N100, P180, and N280 ms [44]. The N40 peak corresponds to the GABA_A_ receptor-based fast IPSP [45,46,47]. Later peaks, including the P60 and N100 peaks, may correspond to the GABA_B_ receptor-based slow IPSP [30,48,49]. Casula and colleagues [50] demonstrated that 1 Hz of rTMS in normal subjects increased the P60 and N100 peaks. Their study was carefully done, but only on 15 subjects and without a placebo-stimulation control group. In this study, we explored whether rTMS at 1 Hz can alter the N40, P60, and N100 peaks, with 47 subjects and two control groups, and we additionally examined cortical sources of the evoked potentials. The goal was to further develop TEPs as a biomarker for assaying regional cortical excitability alterations produced by rTMS in cortical regions that cannot be assayed by peripheral stimulation. This could be useful, for example, for testing excitability in cortical seizure foci or areas of cortical injury.

## 2. Materials and Methods

### 2.1. Subjects

Forty-seven healthy adult participants (age 21–67 years; mea*n* = 32.8 ± 9.6 years) were recruited. Two participants reported left-handedness. All participants signed an informed consent form before participation in present study approved by the Stanford University Institutional Review Board (IRB). Exclusion criteria were extracted from the Rossi et al. (2009) article [51].

### 2.2. TMS Device and Coils

TMS was delivered using an EB Neuro ATES STM9000 magnetic stimulator (EB Neuro S.p.A., Florence, Italy) with the coil held tangentially to the skull, with its handle oriented 45 degrees from midline. TMS was delivered with the electrode cap on, as closely as possible without touching the electrodes. Active TMS was delivered with a 70-mm air-cooled figure-of-eight coil (B9621086004). (EB Neuro S.p.A., Florence, Italy) Pseudo-placebo stimulation was delivered with a visually identical 70-mm air-cooled figure-of-eight coil (B9621086009) (EB Neuro S.p.A., Florence, Italy). The placebo coil stimulated in a tangential plane to cortex, which reduced the effect, but still permitted some stimulation to provide a scalp sensation and mask the treatment group. Therefore, the treatment groups could be considered high versus low stimulation treatment arms, rather than active versus true placebo. A second control group consisting of exclusive use of the placebo coil for 0.20 Hz of TMS and 1 Hz of rTMS blocks was studied to account for possible group differences that might be derived from use of two different coils (Table 1). The group with the active coil for all stimulation was denoted as the full active AAA group (*n* = 22); full placebo as the PPP group (*n* = 10) and the group with the active coil for both sets of 0.20-Hz test TMS and placebo coil for the 1 Hz of rTMS was called the active–placebo–active (APA) group (*n* = 15).

TMS sound artifact was masked via use of white noise played with sound canceling headphones. Volume was incrementally increased until participants reported that the TMS “click” was obscured. Continuous visualization of stimulation site in relation to individual cortical anatomy was ensured using an ATES Medica NetBrain Neuronavigation system. All three test stages followed the same procedure without revealing the coil type to the subjects. Figure 1 shows the experimental arrangement.

### 2.3. EEG System

EEG was recorded using an Electrical Geodesic, Inc. 256-channel MicroCel sensor net. Elefix conductive paste was used on 77 electrodes. Gelled electrodes included the standard 10–20 montage electrodes as well as a denser cluster of electrodes in regions of interest near C3 and C4 (Figure 2). EEG data were recorded referenced to Cz and impedances were kept below 10 kΩ. EEG was sampled at 1 kHz and the amplifier was set to fast recovery. Electrodes were connected to scalp by conductive paste, comprising the 10–20 system and a dense array around C3 and C4, which was the stimulation site.

### 2.4. Resting Motor Threshold (rMT)

Participants were seated in an adjustable chair with a headrest to keep the head stable for the duration of the study. The resting motor threshold (rMT) was determined with the EEG cap in place. The stimulation site was determined by finding location of the stimulation that evoked the largest movement in a participant’s non-dominant hand. rMT was defined as the minimal stimulation intensity used to evoke a visible muscle twitch with time-locked EMG correlate in at least 5 out of 10 trials. When rMT could not be determined (*n* = 6), we set rMT as 65% of maximum stimulator output.

### 2.5. Repetitive Transcranial Magnetic Stimulation (rTMS)

Participants underwent the rTMS procedure in the late morning or early afternoon hours. Participants were asked to keep eyes closed throughout stimulation, but were kept awake for the duration of study, confirmed by EEG and behavioral monitoring. The experiment was limited to a single session that delivered rTMS to non-dominant hand region motor cortex (near the C4 electrode). Previous studies [52,53] provided evidence that non-dominant hand motor cortex and dominant hand motor cortex have similar resting motor thresholds; however, the non-dominant hand motor cortex may be more susceptible to inhibitory stimulation than is the dominant hand motor cortex.

Stimulations were divided into three separate blocks, all delivered with the electrode cap in place. The initial block of rTMS consisted of 200 single pulse rTMS (SpTMS) delivered at 0.20 Hz and 110% of rMT. The second block consisted of 1500 rTMS pulses at 1 Hz and 90% rMT. The rTMS pulses were divided into three sub-blocks of 500 pulses, separated by rest periods of 90–120 s to allow for coil cooling. The final stimulation block consisted of a second round of 200 SpTMS delivered at 0.20 Hz at 110% rMT.

### 2.6. Processing of EEG Data

EEG analysis was performed in MATLAB using EEG-LAB and the TMS-EEG signal analyzer, TESA [54], an open-source extension of EEGLAB. Order of operations for TMS-EEG analysis was the following: (1) EEG data were segmented from −600 ms before to +600 ms after the rTMS pulse. (2) Data were baseline corrected, based on EEG data occurring from −100 to −6 ms. (3) EEG data from −5 to +15 ms were removed and replaced with constant data extrapolated from the pre-artifact baseline, to eliminate the majority of the rTMS pulse artifact. (4) Data were visually inspected for profound artifacts (e.g., flat-lining or noise unrelated to the rTMS). Bad channels and trials were manually removed. (5) TESA performed a first pass of fast independent component analysis (ICA) to correct for rTMS-ringing artifact. (6) EEG data were band-pass filtered from 1–100 Hz and band-stop filtered from 59–61 Hz. (7) A second round of fast ICA was performed to remove remaining artifacts ICA components were grouped by TESA software into one of six categories, including electrode noise, eye-blink, muscle artifact linked to TMS, muscle artifact not linked to TMS, sensory artifact, and other. These were reviewed manually and accepted or rejected based upon topography, being in an isolated topographic island, localization only at sites of muscles or eye movement artifact, frequency spectrum, and waveform shape. When in doubt, potentials were included in the reconstruction. This was not done blinded as to treatment, but treatment was not actively considered during decisions about artifact. (8) Data were re-referenced to an average reference, and data were baseline corrected from −100 to −6 ms. (9) TEPs were averaged across all trials and the mean TEPs were then visualized.

### 2.7. Source Localization

To localize the cortical areas with EEG responses to left motor cortex magnetic stimulation, we reconstructed source activity in a manner comparable with methods used clinically for surgical evaluation of epilepsy patients [55] albeit with a standardized MRI to build the head model. Using trial group averaged pre-treatment TEP EEG signals, the distribution of current density in cortex over time was estimated using low-resolution brain electromagnetic tomography (LORETA) with loose constraints within the Brainstorm plugin for MATLAB. Results from LORETA were similar to those from LAURA (low-resolution electromagnetic tomography) so only results from LORETA were reported. This approximates the optimal current density at the cortical sources needed to produce a distribution of observed EEG potentials over the scalp.

### 2.8. Statistical Analysis

An analysis of covariance (ANCOVA) at α = 0.05 was used to measure differences in TEP peak amplitudes in post-rTMS trials while controlling for pre-rTMS TEP peak amplitude differences. This allowed for adjustment of TEP amplitudes while accounting for any pre-existing differences between groups. If a significant effect was detected by the ANCOVA, Bonferroni-corrected post hoc analyses were conducted to decompose the effect. No ANCOVA assumptions were violated for the reported analyses.

## 3. Results

No clinical or electrographic seizures were induced by the stimulation. Except for occasional mild scalp discomfort, all were able to tolerate the procedure.

### 3.1. Motor-Evoked Potential (MEP)

The 47 subjects demonstrated a resting motor threshold of 67.47 ± 7.05 % of maximum machine output. There was no significant change in motor evoked potential (MEP) after 1 Hz of rTMS (*p* = 0.26, Cohen’s d = 0.17). The MEP amplitude was not correlated significantly with the P30, N40 (Pearson r = −0.04, *p* = 0.43) or N100 (Pearson r = 0.32, *p* = 0.07) peaks or changes in peak amplitudes after rTMS (Pearson r = −0.01, *p* = 0.49 for N40 and Pearson r = 0.10, *p* = 0.39 for N100).

### 3.2. TEP Latencies and Amplitudes

Within individual subjects, TEP peak latencies were reliable with variations between pre- and post-rTMS treatments of no more than ±4.18 ms. Between subjects, the peak latencies however showed substantial variability, with various individuals showing increases, decreases, or no change from pre -rTMS to post- rTMS.

Mean amplitudes (see Figure 3) for the test block TEPs in the fully active (AAA) group before and after 1 Hz of rTMS show that P60 and N100 amplitudes both became more negative (P60 amplitude decreased and N100 increased). The mixed (APA) group demonstrated increased (more negative) N40 amplitudes after 1 Hz of rTMS. The full placebo stimulation (PPP) group did not consistently have well-formed TEP components.

Full placebo stimulation (PPP) produced poorly formed low-amplitude (see Table 2) waveforms. Table 2 indicates that the amplitudes of the TEPs to the 0.2-Hz test pulses were generally similar before and after 1-Hz treatments, but the absolute amplitude of the all-placebo response was about 40% of those evoked by active stimulation. To determine the effect of active 1-Hz rTMS, independent of effects of the baseline amplitude of TEPs and of placebo stimulation, we performed ANCOVAs among the experimental conditions on post-rTMS amplitudes, while controlling for pre-rTMS TEP peak amplitudes, with post hoc testing conducted to decompose significant ANCOVAs (Figure 4).

### 3.3. N40

The baseline-corrected N40 peak amplitude increased in the AAA group after rTMS, but the change did not achieve significance according to the one-way analysis of covariance (ANCOVA), F(2, 34) = 1.284, ηp2 = 0.070, 1–β = 0.259, and *p* = 0.290. The N40 peak did not show a significant difference among groups, although the AAA group was descriptively larger after active stimulation (mean adjusted = −1.174, standard error (SE) = 0.426) compared to PPP (mean adjusted = 0.042, SE = 0.681), d = 0.676, and APA (mean adjusted = −0.493, SE = 0.500) d = 0.378 groups. The amplitude of APA was slightly more negative than PPP, d = 0.297.

### 3.4. P60

The amplitude-corrected P60 peak became less positive after active rTMS for the AAA group compared to the APA group. A one-way analysis of covariance (ANCOVA) of pre- versus post-TEP P60 amplitudes across the three experimental conditions, while controlling for pre-rTMS TEP P60 amplitudes revealed a significant effect of experimental group, F(2, 39) = 5.494, ηp2 = 0.220, 1–β = 0.822, and *p* = 0.008. Bonferroni-adjusted post hoc analyses showed a significant difference between the AAA condition (mean adjusted = 1.287, SE = 0.244), t(39) = 2.889, *p* = 0.024, d = 0.996 and the APA condition (mean adjusted = 2.398, SE = 0.308), There was also a significant difference between the PPP condition and the APA condition (mean adjusted = 0.860, SE = 0.403), t(39) = 3.113, *p* = 0.016, d = 0.1380. There was not a significant difference between the AAA and PPP conditions, t(39) = 0.922, *p* = 1.000, d = 0.383. Although the PPP group demonstrated a significantly more positive P60 peak amplitude after 1 Hz of rTMS in the averaged raw waveforms with the placebo coil, the placebo coil had a high degree of variation, obscuring potentially significant differences.

### 3.5. N100

The N100 peak increased with 1 Hz of rTMS in the AAA group. ANCOVA was conducted across the three experimental conditions on post-rTMS TEP N100 amplitudes controlling for pre-rTMS TEP N100 amplitudes. The analysis revealed a significant effect of experimental group, F(2, 40) = 3.295, ηp2 = 0.141, 1–β = 0.593, and *p* = 0.047, with N100 descriptively more negative amplitudes in the AAA group compared to the other groups after rTMS. However, Bonferroni-adjusted post hoc analyses failed to show significant differences between the APA condition (mean adjusted = −1.373, SE = 0.665), the AAA condition (mean adjusted = −3.262, SE = 0.565), and the PPP condition (mean adjusted = −1.041, SE = 0.921), all t-values ≤ 2.169, all *p*-values ≥ 0.109.

### 3.6. Topography of TEPs

rTMS caused topographically widespread TEP components. The early TEP waveforms between 10 and 30 ms had larger amplitudes at electrode locations near C4, but later components, while being visible contralaterally from the stimulation site, had their highest amplitude peaks in electrodes covering the stimulated cortical areas. Later waveforms such as the N100 and P180 were characterized by more profound bilateral distribution compared to earlier waveforms. When comparing TEP amplitudes pre-post changes at electrode sites distant from the stimulation site we found no significant changes in any groups (Figure 5).

### 3.7. Source Analysis

The localized source activity for TEPs were shown to be likely evoked from focal areas near or under the site of stimulation. However, later waveforms (N100 and P180) affected a larger cortical area. At the initial stimulation, source activity shows a wide area effected by rTMS near the site of stimulation around the C4 electrode, including the ipsilateral precentral gyrus, superior frontal gyrus, and middle frontal gyrus, and to a lesser extent the ipsilateral postcentral gyrus and contralateral superior precentral and postcentral gyri (Figure 6). Source analysis of the P30 and N40 waveforms show activity generators predominantly anterior to the site of rTMS including the ipsilateral middle frontal and superior frontal gyri. The P60 waveform shows activity generators in the ipsilateral precentral, postcentral, supramarginal, and superior frontal gyri. The N100 waveform shows widespread activation, including generators in the temporal poles, frontal poles, superior frontal gyri, superior parietal lobes, ipsilateral pre- and post-central gyri, superior portions of the contralateral pre- and post-central gyri, and the ipsilateral middle frontal gyrus. The P180 waveform generators arise predominantly from the superior parietal lobes, ipsilateral superior frontal, middle frontal gyri, and the contralateral superior frontal lobe and, to a lesser extent, the contralateral pre- and post-central gyri, and the superior and middle temporal gyri.

## 4. Discussion

This study in normal volunteers confirms that 1 Hz of repetitive transcranial magnetic stimulation alters rTMS-evoked EEG potential (TEP) waveforms. These waveforms can then be rendered visible with signal averaging and processing [24,26,28,30,44,56]. Numerous studies [57,58] have evaluated short-interval intracortical inhibition (SICI, typically 1–5 ms) and long-interval intracortical inhibition (LICI, typically 50–200 ms) by measuring the EEG response to paired TMS pulses. Our study evaluated the effect of 1-Hz repetitive pulse trains on TEP waveforms, which is a much less commonly employed experimental paradigm than is paired-pulse stimulation, but one that might have greater potential for evaluation of different regions of cortex.

Previous related work includes a study by Casula and associates who found that rTMS increased the P60 and N100, but not the P30 or N40 [50]. That study did not control for pre-existing baseline amplitude differences and did not use a placebo comparison group. We observed a less positive (smaller) P60 and more negative (bigger) N100 after 1 Hz of rTMS. This partially confirmed the results found by Casula, while controlling for the confounding factor of highly variable TEP amplitudes among different subjects. The divergent findings of our study versus those of Casula regarding the P60 may be in part due to differences in our procedures. Casula and colleagues used 120% RMT and 50 single pulses to measure TEP amplitudes. We used 110% RMT and 200 single pulses to measure TEP amplitudes, which might have improved the signal-to-noise ratio. Additionally, the present study evaluated a larger study population, with a different method of marking peak amplitudes, and addition of placebo comparison groups.

A control group with placebo stimulation is important to rule out nonspecific effects, because rTMS produces auditory and somatosensory components of the TEP that are not directly related to magnetic stimulation-induced cortical activity [59]. We were able to show group difference for several peaks generated by test stimulations at 0.20 Hz for placebo or active coils before and after 1 Hz of rTMS. Our placebo coil stimulation was able to evoke variable and low amplitude cortically-generated waveforms; therefore, our placebo stimulations might better be considered a low-dose comparator to active stimulation, meaning that comparisons of active to placebo stimulation might have underestimated the effects of active stimulation.

Only limited information is available about the physiological relevance of TEP peaks. Pharmacological studies suggest that the N40 is enhanced in humans with administration of diazepam, reflecting increased fast GABA_A_ mediated inhibition [45,46,47,59,60]. The N100 peak increases with the GABA_B_ agonist, baclofen [45,48] and decreases with presynaptic inhibitors of excitatory transmitter release [61] implying potential for serving as a marker of cortical inhibition that might be useful for reducing seizures [30]. The amplitude of the N100 peak might depend more on the ratio of GABA to glutamate than upon GABA alone [60]. However, increase in GABA_B_-mediated synaptic inhibition might be expected to produce variable effects in different types of epilepsy. Absence seizures, for example, show spike-wave EEG discharges, with the wave component reflecting a significant component of slow GABA_B_-mediated inhibitory potentials [62,63].

The P60 peak occurs at a time of long-interval intracortical inhibition, also mediated by GABA_B_ receptors [45]. Excitatory transmission may play a role in generating the P60 peak, since the glutamate AMPA receptor antagonist, perampanel, suppresses P60 amplitude [61,64]. Rogasch [30] suggested that the P60 TEP could reflect a component of somatosensory feedback, but Cash [49] has argued that there is a significant cortical component of P60.

Identifying cortical sources of TEPs could be important for rTMS use in a clinical setting. Our study examined TEPs averaged across participants and modeled sources coming from a normalized atlas brain. The TEP activity localized near the stimulation site, but also with some distant activity, reflecting network spread. We again confirm the findings of Casula [50] and Bikmullina [65] that motor cortex rTMS influences TEPs over a wide region of ipsilateral cortex. In our study, the range of effect was from anterior temporal to occipital regions. Studies using motor assays have identified transcallosal inhibition provoked by contralateral motor cortex rTMS [66,67] and we can confirm a significant contralateral component of the TEPs. The widespread effects of rTMS to increase the N40 marker on fast inhibition might argue against the need for exact targeting of stimulation. However, enhancement of inhibition was maximal at the motor cortex stimulation site. TEP changes in this study did not correlate with amplitude of the motor evoked responses, as has been noted by others [28]. We did not systematically look for changes in motor threshold or paired-pulse inhibition at various inter-stimulus intervals. Studies using individual MRI brain modeling could take advantage of rTMS with cortical source modeling to create detailed individual connectivity maps by plotting propagation patterns of TEPs evoked from systematically chosen cortical areas.

Variability among participant’s TEP responses is high [68], limiting the significance of post hoc pair-wise and group comparisons. However, individuals have relatively stable TEPs at 20, 30, 40, 60, and 100 ms [69], suggesting that TEPs might provide a useful biomarker for regional cortical excitability in specific patients. Of the TEPs we found the N100 to be the most viable biomarker for rTMS induced cortical inhibition. If TEPs are confirmed as a reliable surrogate marker for cortical inhibition in epilepsy patients, then TEP recordings could become significantly more efficient than seizure counts in screening the effects of anti-seizure therapies. Of course, any such findings would require validating the effect of a possible treatment on seizure counts.

Our study is subject to several interpretive limitations. Distinguishing continuous waveforms is problematic, because changes in adjacent peaks can be additive or subtractive, rather than independent. For this reason, we employed unbiased software measurements of amplitudes at selected peaks, rather than trough-to-peak values, but this may have been at the expense of evaluating effects on individual peaks. TMS-TEPs are subject to a wide variety of artifacts [70]. Late TEP components can possibly be influenced by improper masking of the loud rTMS “click” that occurs during stimulation [71]. We attempted to account for this by use of white noise played via sound cancelling earbuds, but we cannot guarantee that the rTMS “click” was completely obscured. We have no evidence that the changes in TEP waveforms correlate with any clinical or even biological effect. Examining grand averaged source space activity likely misses the nuances that would be critical to clinical application of rTMS source reconstruction. rTMS-evoked EEG potentials might differ from normal volunteers versus those with epilepsy or other neurological diseases when stimulating the areas of neurological abnormality. Our experiments do not document the durability and replicability of TEP changes. Future clarification of these issues will further the use of rTMS-evoked EEG potentials as biomarkers for cortical excitability in non-motor cortex.

## 5. Conclusions

As described by several prior studies, rTMS evokes measurable EEG potentials, discernible after suitable processing. A positive peak at 60 ms and a negative peak at 100 ms are each altered by 1 Hz of repetitive TMS at motor cortex, with P60 decreased and N100 increased. N40 showed a non-significant trend towards an increase. Effects on other evoked peaks are variable. While several of our findings are confirmatory of previous work, new features include reliable persistence of evoked EEG potential changes in response to rTMS when controlling for highly variable initial amplitude and as compared to sham-stimulation controls. Comparing changes in the EEG, not only to before-after rTMS, but to sham stimulation controls confirm the important increase of the N100 potential, but are less confirmatory of P60 peak changes. Our topographic dipole analysis documents that the largest effects of 1-Hz rTMS on TEPs are manifest early and close to the stimulation site. These changes in rTMS-evoked EEGs may reflect increased GABA_B_-mediated inhibition in specific brain regions.

Limitations of this technology are several, including difficulties in isolating individual EEG peaks and intra-subject variability, often requiring population averages and statistics to visualize changes. However, measurements of rTMS-evoked EEG changes are not restricted to motor cortex, so they may serve as useful biomarkers for cortical excitability at a seizure focus.

## Figures and Tables

**Figure 1 sensors-22-01762-f001:**
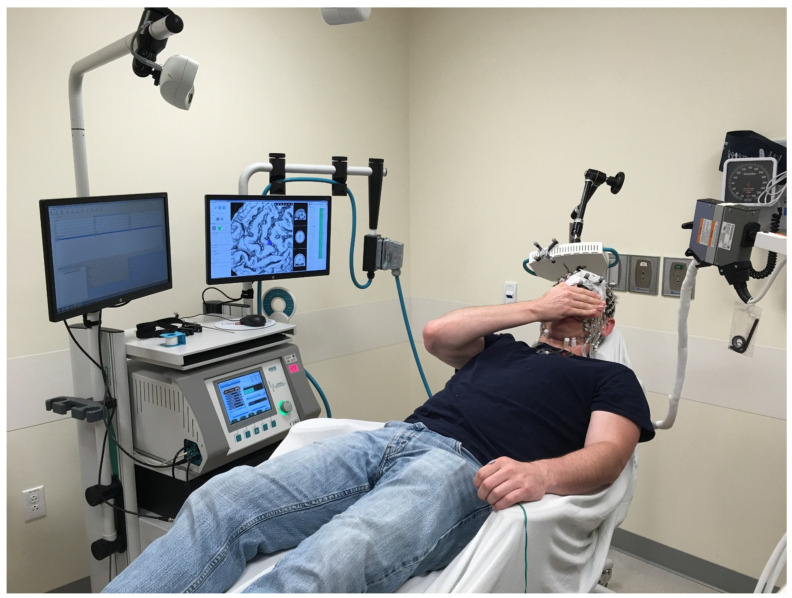
rTMS setup showing the subject in relation to the stimulator generator and coil, neuronavigation system, wire to the left thumb to stimulate and record EMG and (to the right) cable to the EEG machine.

**Figure 2 sensors-22-01762-f002:**
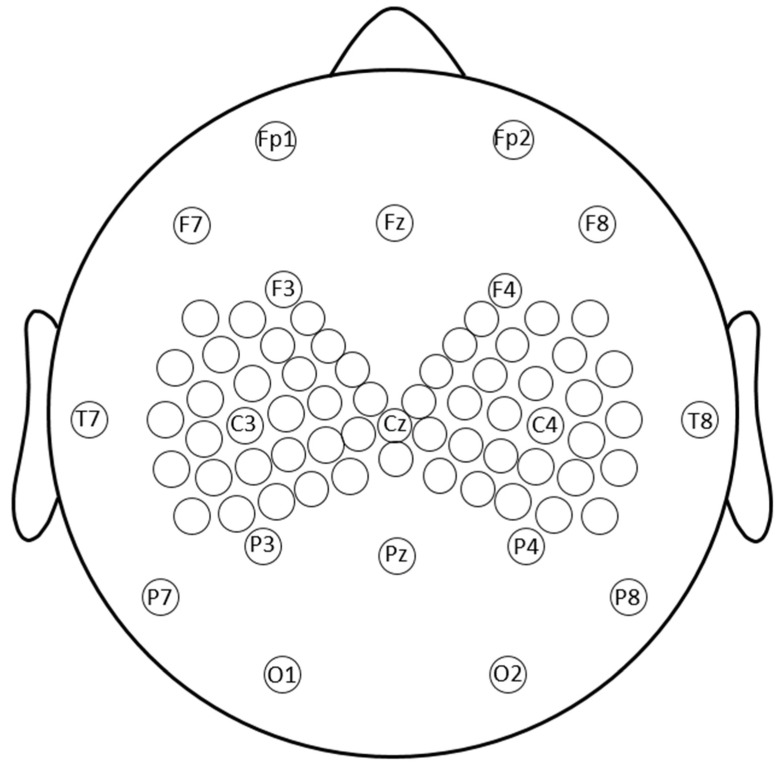
Topography of the recording electrodes. EEG was recorded using an Electrical Geodesic, Inc. 256-channel MicroCel sensor net. All electrodes on the sensor net were spaced 0.5–1.5 cm from each other at the center. Conductive paste was used on 76 electrodes including the standard 10–20 montage electrodes and a denser cluster of 27 electrodes near the TMS region of interest, collectively falling within 6 cm of C3 and C4, respectively.

**Figure 3 sensors-22-01762-f003:**
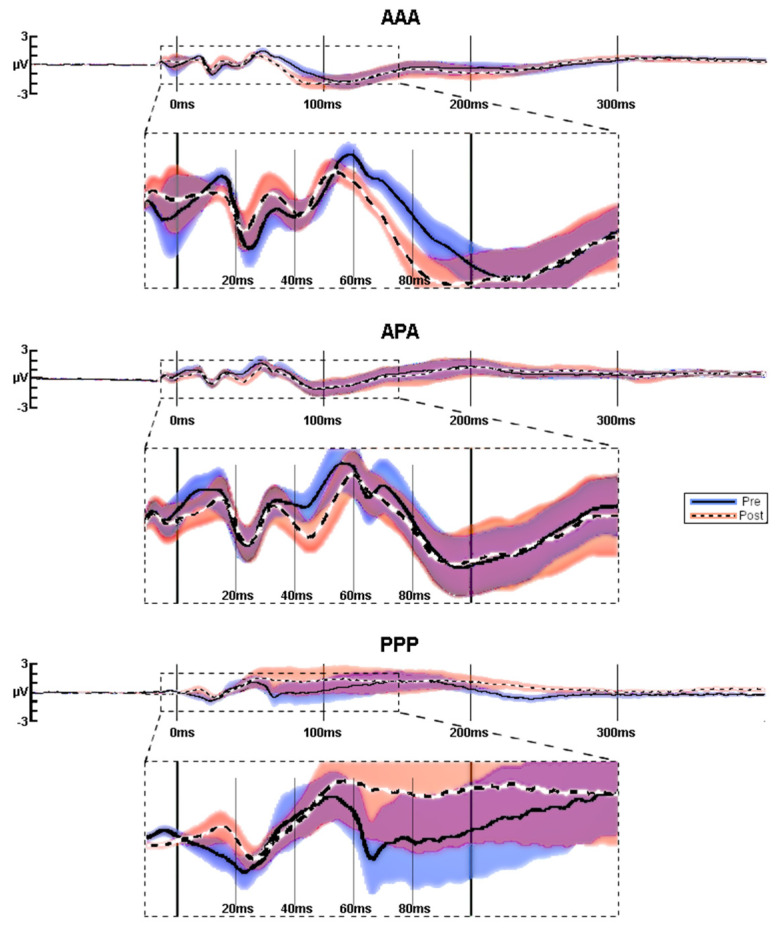
TEPs averaged at C4. Average pre (blue) vs. post (red) TEPs in response to 1-Hz rTMS for AAA (*n* = 22), APA (*n* = 15), and PPP (*n* = 10) groups. Since there was jitter in peak latencies between participants, the amplitudes of the grand averaged traces are lower than they would have been, had each peak been adjusted individual for peak latency (see Figure 3). Significance was not calculated for these raw averaged amplitudes, but on baseline-corrected peak amplitudes.

**Figure 4 sensors-22-01762-f004:**
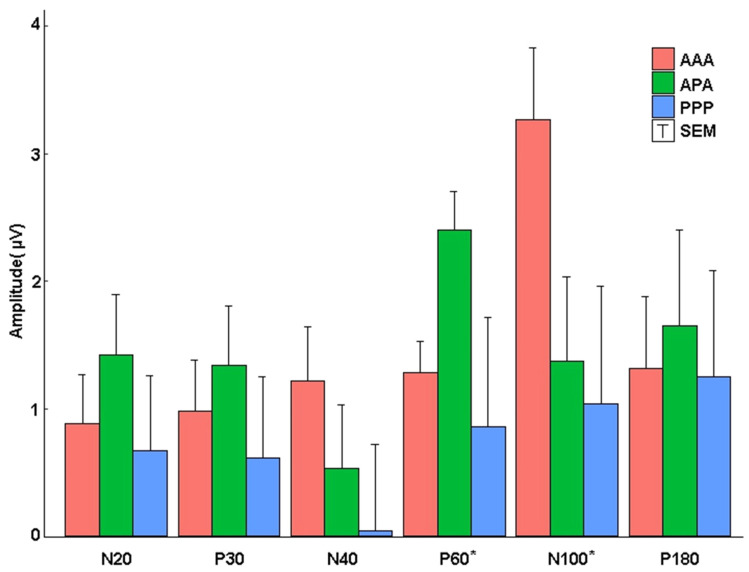
Absolute ANCOVA adjusted peak amplitudes. The amplitude of post-corrected for pre-rTMS TEP peaks for each experimental group. No significant changes before and after active 1-Hz rTMS were noted for N20 and P30. AAA showed a reduced P60 amplitude post rTMS compared to the APA stimulation group. AAA showed a more negative N100 amplitude after rTMS when compared to APA and PPP groups. (* *p* < 0.05). T-bars represent half the standard error.

**Figure 5 sensors-22-01762-f005:**
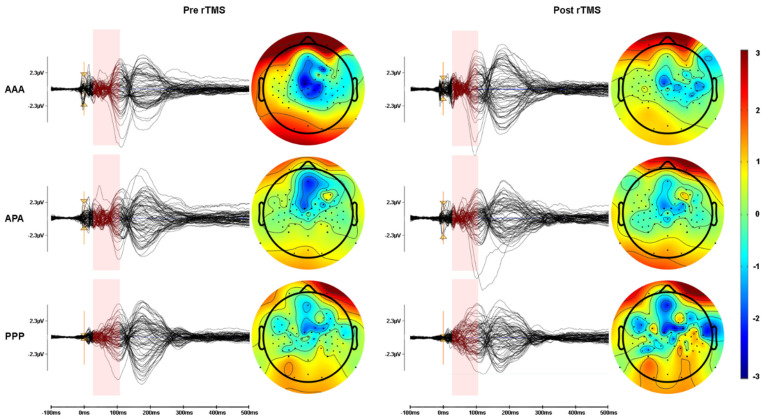
Group average TEP topography. Butterfly plot of EEG traces and topographic portrayal of TEP amplitude averaged from 20–100 ms in response to the stimulation of left motor cortex region. Left is left in the figure and right is right.

**Figure 6 sensors-22-01762-f006:**
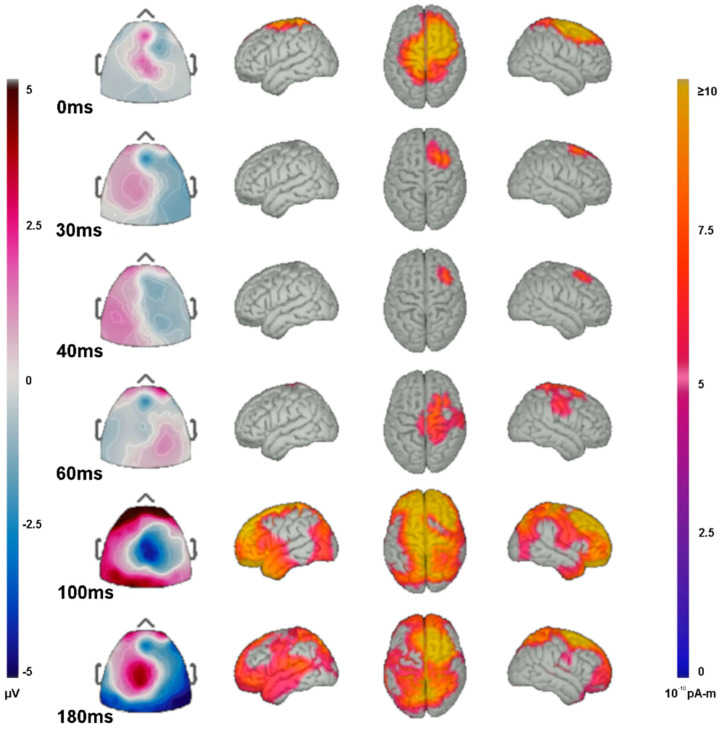
Source localization of TEP activity. Butterfly plot of EEG traces and topographic portrayal of TEP amplitude averaged from 20–100 ms in response to the stimulation of left motor cortex region. Right is on the left (MRI convention) and left is on the right.

**Table 1 sensors-22-01762-t001:** Test stages.

Group	*n*	Test Pre 0.20 Hz	rTMS 1 Hz	Test Post 0.20 Hz
Full active (AAA)	22	active	active	active
Full placebo (PPP)	10	placebo	placebo	placebo
Mixed (APA)	15	active	placebo	active

**Table 2 sensors-22-01762-t002:** Baseline-adjusted amplitudes. All units are µV. SUM of AV represents the sum of the absolute values of the peak amplitudes and their standard error.

	AAA	PPP	APA
N20	−0.89 ± 0.38	−0.68 ± 0.59	−1.42 ± 0.48
P30	0.99 ± 0.39	0.62 ± 0.64	1.35 ± 0.46
N40	−1.17 ± 0.43	0.04 ± 0.68	−0.49 ± 0.50
P60 *	1.29 ± 0.24	0.86 ± 0.86	2.40 ± 0.31
N100 *	−3.26 ± 0.57	−1.04 ± 0.92	−1.37 ± 0.67
P180	1.32 ±0.57	1.25 ± 0.84	1.65 ± 0.75
SUM of AV	8.91	3.48	8.68

## Data Availability

Data may be provided upon request to Mr. Adam Fogarty at afogarty@stanfordhealthcare.org, depending upon circumstances of the request.

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
