# Peer review of "EEG Evoked Potentials to Repetitive Transcranial Magnetic Stimulation in Normal Volunteers: Inhibitory TMS EEG Evoked Potentials"

_sensors, 2022, doi:10.3390/s22051762_

Round 1

Reviewer 1 Report

This paper is targeting a relevant research area and as such will surely be of interest of Readers.

There was a convincing rationale provided for the research. All experiments are well planned, executed and results nicely presented and discussed at the sufficient level of detail. 

As of a bit of criticism, the biggest problem I have with the paper is that there is no clearly stated / emphasised contribution of the paper to the research area. Authors state in the conclusion that this paper is one of many in this very research domain but from reading this paper there is really no clarity as to what is the specific contribution of this paper. Such a thing should be included either, in the abstract or in the related work section where Authors could, while providing related work, refer their work to what others have done. The thing is that there is no Related work section in this paper. Even if the Introduction partially serves as one, the purpose of Introduction section and the Related work section is completely different hence it may be worth to leave the rational and stating the state of the art in the Introduction section while move discussions over what others have done in this domain to a separate Related work section> This is just my suggestion and not a must but in my view the paper would benefit from a thoroughly composed Related work section. 

The next thing is conclusion - it is very short, one paragraph and six sentences raises question what is the purpose? I understand that what typically is being included in the conclusion section in this paper is included in the Discussion section preceding the Conclusion section. And even if in my opinion the discussion is very much on-point and valuable, maybe it would be possible to extract something from the Discussion section and include this in the Conclusion section to make the latter more meaningful? Again, not a must but a suggestion.

Other than that, references are fine, so is the language.

And last but not least - after reviewing hundreds of papers I have never seen before degrees / education indicated directly in the Authors list which usually contains names only.

Author Response

Reviewer 1

This paper is targeting a relevant research area and as such will surely be of interest of Readers.

There was a convincing rationale provided for the research. All experiments are well planned, executed and results nicely presented and discussed at the sufficient level of detail. As of a bit of criticism, the biggest problem I have with the paper is that there is no clearly stated / emphasised contribution of the paper to the research area. Authors state in the conclusion that this paper is one of many in this very research domain but from reading this paper there is really no clarity as to what is the specific contribution of this paper. Such a thing should be included either, in the abstract or in the related work section where Authors could, while providing related work, refer their work to what others have done.

  1. NEW CONTRIBUTION: Good point. We put a sentence up front in the abstract: “New features are controls for baseline amplitude and two types of control groups.” And we also say this in the discussion.

------------------------------------------------------

  1. NO RELATED WORK SECTION: There is no Related work section in this paper. Even if the Introduction partially serves as one, the purpose of Introduction section and the Related work section is completely different hence it may be worth to leave the rational and stating the state of the art in the Introduction section while move discussions over what others have done in this domain to a separate Related work section. This is just my suggestion and not a must but in my view the paper would benefit from a thoroughly composed Related work section. 

The main related work is the important study by Casula and we highlighted this by calling it “Previously related work.”

------------------------------------------------------

3.SHORT CONCLUSION: The next thing is conclusion - it is very short, one paragraph and six sentences raises question what is the purpose? I understand that what typically is being included in the conclusion section in this paper is included in the Discussion section preceding the Conclusion section. And even if in my opinion the discussion is very much on-point and valuable, maybe it would be possible to extract something from the Discussion section and include this in the Conclusion section to make the latter more meaningful? Again, not a must but a suggestion.

Good suggestion - we expanded the Conclusions.

------------------------------------------------------

  1. GOOD LANGUAGE: Other than that, references are fine, so is the language.

Thank you.

------------------------------------------------------

  1. TAKE OUT AUTHOR DEGREES: And last but not least - after reviewing hundreds of papers I have never seen before degrees / education indicated directly in the Authors list which usually contains names only.

Removed

Reviewer 2 Report

The authors describe experiments with repetitive transcranial magnetic stimulation and detection of EEG evoked potentials. The topic is interesting. However, the motivation for such type of experiments should be explained.

There are more items in the text that need clarification and/or addition of more information.

Section 2:

Please add a photo with magnetic stimulator positioned at the head.

For EEG there is written that "a denser cluster of electrodes in regions of interest near C3 and C4" were used. Please add a figure showing precise location of these electrodes.

In the next paragraph it is mentioned that the stimulation site was determined by finding location that evoked the largest movement in the non-dominant had. PLease explain why the non-dominant hand was used and not the dominant hand. This is a medical knowledge that mus be explained to readers.

The sub-section Processing of EEG Data is described rather superficially and not very clearly. In particular the lines 152-154 need more detailed description - how the authors determined what is useful signal and what is an artifact. I assume it was not only by visual inspection but some methods were used. These methods must be described.

Section 3:

Figures 2, 3, 4 are not referenced in the text.

Line 257: "electrode locations near C3" - please specify more precisely or show in a figure - how many electrodes, which distances

Section 5 Conclusions

This section is very brief. One would expect that outline of future research in the area is proposed.

Reference 1 is incomplete. There is no author and no title of the article.

Author Response

Reviewer 2

The authors describe experiments with repetitive transcranial magnetic stimulation and detection of EEG evoked potentials. The topic is interesting. However,

6.EXPLAIN MOTIVATION: the motivation for such type of experiments should be explained.

Understood. We expanded the last sentence of the introduction to document our motivation: “The goal is to further develop TEPs as a biomarker for assaying regional cortical excitability alterations produced by rTMS in cortical regions that cannot be assayed by peripheral stimulation. This could be useful, for example, for testing excitability in cortical seizure foci or areas of cortical injury.”

------------------------------------------------------

There are more items in the text that need clarification and/or addition of more information.

  1. ADD MAGNETIC STIMULATOR PHOTO: Section 2: Please add a photo with magnetic stimulator positioned at the head.

Done

Legend: rTMS setup showing the subject in relation to the stimulator generator and coil, neuronavigation system, wire to the left thumb to stimulate and record EMG and (to the right) cable to the EEG machine.

------------------------------------------------------

  1. FIGURE SHOWING ELECTRODE CLUSTER AROUND C3 & C4: For EEG there is written that "a denser cluster of electrodes in regions of interest near C3 and C4" were used. Please add a figure showing precise location of these electrodes.

Done, see figure below

Legend: Topography of the recording electrodes. EEG was recorded using an Electrical Geodesic, Inc. 256-channel MicroCel sensor net. All electrodes on the sensor net were spaced 0.5-1.5 cm from each other at the center. Conductive paste was used on 76 electrodes including the standard 10-20 montage electrodes and a denser cluster of 27 electrodes near the TMS region of interest, collectively falling within 6 cm of C3 and C4, respectively.

------------------------------------------------------

9.WHY NON-DOMINANT HAND: In the next paragraph it is mentioned that the stimulation site was determined by finding location that evoked the largest movement in the non-dominant hand. Please explain why the non-dominant hand was used and not the dominant hand. This is a medical knowledge that must be explained to readers.

Previous studies (Ridding and Flavel 2006, Poole et al. 2018) have shown evidence that non-dominant hand motor cortex and dominant hand motor cortex have similar resting motor threshholds (rMT), however non-dominant hand motor cortex may be more susceptible to inhibitory stimulation than dominant hand motor cortex.

Ridding, M. C., & Flavel, S. C. (2006). Induction of plasticity in the dominant and non-dominant motor cortices of humans. Experimental brain research, 171(4), 551–557. https://doi.org/10.1007/s00221-005-0309-2

Poole, B. J., Mather, M., Livesey, E. J., Harris, I. M., & Harris, J. A. (2018). Motor-evoked potentials reveal functional differences between dominant and non-dominant motor cortices during response preparation. Cortex; a journal devoted to the study of the nervous system and behavior, 103, 1–12. https://doi.org/10.1016/j.cortex.2018.02.004

------------------------------------------------------

  1. BETTER EXPLAIN HOW ARTIFACT WAS EXCLUDED IN ICA: The sub-section Processing of EEG Data is described rather superficially and not very clearly. In particular the lines 152-154 need more detailed description - how the authors determined what is useful signal and what is an artifact. I assume it was not only by visual inspection but some methods were used. These methods must be described.

ICA components were grouped by TESA software into one of six categories, including electrode noise, eye-blink, muscle artifact linked to TMS, muscle artifact not linked to TMS, sensory artifact and other. These were reviewed manually and accepted or rejected based upon topography, being in an isolated topographic island, localization only at sites of muscles or eye movement artifact, frequency spectrum and waveform shape. When in doubt, potentials were included in the reconstruction. This was not done blinded as to treatment, but treatment was not actively considered during decisions about artifact.

------------------------------------------------------

  1. MAKE REFERENCE IN TEXT TO FIGURES 2, 3 and 4.

Done, including new figures

------------------------------------------------------

  1. In LINE 257, MORE DETAIL ON "electrode locations near C3" - please specify more precisely or show in a figure - how many electrodes, which distances

Please see the figure

------------------------------------------------------

  1. SHORT CONCLUSION SECTION: This section is very brief. One would expect that outline of future research in the area is proposed.

Agreed - Conclusion has been expanded

------------------------------------------------------

  1. INCOMPLETE REFERENCE 1: Reference 1 is incomplete. There is no author and no title of the article.

Thanks for noticing this typo. The full reference was inserted. Barker AT, Jalinous R, Freeston IL. Non-invasive magnetic stimulation of human motor cortex. Lancet. 1985 May 11;1(8437):1106-7. doi: 10.1016/s0140-6736(85)92413-4.

Round 2

Reviewer 2 Report

The manuscript has been improved. The authors responded to comments and questions appropriately.